# Segment anything small for ultrasound: Enhancing segmentation with non-generative augmentation

Danielle L. Ferreira⊙*, Ahana Gangopadhyay, Hsi-Ming Chang, Ravi Soni, Gopal Avinash

Science and Technology Office, GE HealthCare, San Ramon, California, United States of America

* llopes.danielle@gmail.com

## Abstract

Accurate segmentation of anatomical structures in ultrasound (US) images, particularly small ones, is challenging due to noise and variability in imaging conditions (e.g., probe position, patient anatomy, tissue characteristics and pathology). To address this, we introduce Segment Anything Small (SAS), a simple yet effective scale- and texture-aware data augmentation technique designed to enhance the performance of deep learning models for segmenting small anatomical structures in ultrasound images. SAS employs a dual transformation strategy: (1) simulating diverse organ scales by resizing and embedding organ thumbnails into a black background, and (2) injecting noise into regions of interest to simulate varying tissue textures. These transformations generate realistic and diverse training data without introducing hallucinations or artifacts, improving the model's robustness to noise and variability. We fine-tuned a promptable foundation model on a controlled organ-specific medical imaging dataset and evaluated its performance on one internal and five external datasets. Experimental results demonstrate significant improvements in segmentation performance, with Dice score gains of up to 0.35 and an average improvement of 0.16 [95% CI: 0.132,0.188]. Additionally, our iterative point prompts provide precise control and adaptive refinement, achieving performance comparable to bounding box prompts with just two points. SAS enhances model robustness and generalizability across diverse anatomical structures and imaging conditions, particularly for small structures, without compromising the accuracy of larger ones. By offering a computationally efficient solution that eliminates the need for extensive human labeling efforts, SAS enables performance improvements in medical image analysis, particularly in resource-constrained settings.

**Data availability statement:** The datasets used in this study are summarized in Table 1 and include both publicly available and internal datasets. Public datasets used for performance evaluation are openly accessible and are referenced in Table 1. Additional datasets, including Kidney, Gynecology, and Follicle, are internal, non-public datasets and are not publicly available due to protected patient health information and institutional data governance restrictions. These datasets were used for model training (Kidney and Gynecology) and evaluation (Follicle). Access to these data may be considered upon reasonable request to the HealthCare Real World Data (RWD) Team at: rwd@gehealthcare.com, subject to institutional approval and applicable data-sharing agreements.

**Funding:** The author(s) received no specific funding for this work.

**Competing interests:** I have read the journal's policy, and the authors of this manuscript have the following competing interests: all authors were employed by GE Healthcare during the research and manuscript preparation. However, the research was conducted independently, and the authors declare that the results and conclusions are presented objectively, without external influence from their employer.

## Author summary

We developed a simple way to help artificial intelligence better outline structures in ultrasound images, especially very small ones that are often difficult to detect accurately. Ultrasound images can vary a lot because of differences in body shape, probe position, image quality, and disease. These changes make it harder for computer models to work reliably across patients and settings. In our study, we created a training approach that makes existing ultrasound images appear at different sizes and with different texture patterns, while still preserving realistic anatomy. This gave the model more varied examples to learn from without creating artificial structures or requiring large amounts of new manual labeling. We found that this approach improved performance across several ultrasound datasets, including challenging cases involving small targets such as follicles, breast tumors, and thyroid nodules. The benefits were strongest when training data were limited, but improvements remained even with much larger datasets. We believe this work shows that carefully designed image transformations can make ultrasound analysis more accurate and robust, which could help support medical imaging tools in settings where expert-labeled data are scarce.

## Introduction

Accurate multi-organ segmentation is critical for applications such as computer-aided diagnosis, surgical planning, navigation, and radiotherapy. However, the task presents significant challenges given the wide variation in organ sizes, ranging from large structures in the chest and abdomen to smaller ones in the head, neck, and gynecological regions. These size disparities often lead to poor segmentation performance, particularly for smaller organs. This, along with high variability of the background region can result in ambiguity in discriminating structure boundaries, followed by the subsequent deterioration of the segmentation performance [1].

Previous work has experimented with several approaches to improve small organ segmentation performance in the context of expert models that were fine-tuned for segmenting specific organs. Cascaded multi-stage networks have often been used for coarse-to-fine segmentation to iteratively shrink the input region, leading to more accurate segmentation of small anatomical structures [2–4]. Other approaches include incorporating attention mechanisms in U-Net upsampling layers [5], using additional sub-networks for localization [6], imposing shape constraints [6], or employing multi-level structural loss functions that fuse region boundary and pixel-wise information [7]. However, these methods often require architectural modifications and introduce additional computational overhead. Moreover, many are optimized for specific organs, limiting their generalizability.

In contrast, data augmentation provides a model- and anatomy-agnostic alternative to enhancing segmentation performance. Yet, conventional augmentation strategies, such as geometric transformations and intensity perturbations, do not always

enhance performance for small organ segmentation, as they may introduce unrealistic variations or fail to sufficiently expand the representation of small structures in training data [8,9]. This highlights the need for specialized augmentation techniques tailored to small organ segmentation in ultrasound imaging.

More recently, foundation models trained on large-scale, diverse datasets have demonstrated remarkable success across multiple domains. These models serve as adaptable backbones for various downstream tasks through fine-tuning. The Segment Anything Model (SAM) [10], for instance, has been fine-tuned on 2D medical images, enabling prompt-based segmentation across different modalities and anatomies [11,12]. However, while these models perform well on general segmentation tasks, small organ segmentation remains an area with significant room for improvement [12,13].

In this work, we address these challenges by introducing Segment Anything Small (SAS), a dual transformation strategy designed to enhance data variability in terms of organ scale and tissue texture. Unlike conventional augmentation techniques, SAS explicitly improves the representation of small structures, making it particularly well-suited for multi-organ ultrasound segmentation. To evaluate its effectiveness, we fine-tuned a lightweight transformer-based promptable foundation model using SAS in two scenarios: a limited single-organ dataset with low diversity and a larger, more diverse multi-organ dataset. Our experimental results, conducted on one internal and five external datasets from diverse organs, demonstrate that SAS significantly improves segmentation performance across both large and small anatomical structures using click prompts. Notably, our results show that fine-tuning medical imaging foundation models with SAS leads to more robust segmentation across varying organ sizes, even when the fine-tuning dataset is relatively small and lacks diversity. Furthermore, SAS remains effective even as the training dataset expands 80-fold (from 1,050 to 80,000 images), demonstrating its robustness in both data-scarce and data-abundant scenarios.

In summary, this work makes the following contributions:

- We propose SAS, a simple yet effective dual transformation strategy that combines image scaling into thumbnails with region-of-interest (ROI) perturbation, enhancing the network's ability to learn shape and texture information of small organs.

- SAS is a non-generative data variety approach that generates realistic and semantically consistent perturbed images, circumventing regulatory and compliance concerns associated with generative synthetic data approaches, such as those related to the Health Insurance Portability and Accountability Act (HIPAA) and the General Data Protection Regulation (GDPR), as highlighted in [14]. This makes SAS a practical and ethically sound solution for medical imaging. Unlike generative methods, SAS prevents the introduction of artifacts, unwanted structures, or hallucinations, making it highly suitable for aiding in the development of AI-based medical applications, where accuracy and reliability are critical.

- SAS enhances small-organ segmentation performance while maintaining accuracy for larger organs. Through extensive experiments on both small and large organ segmentation tasks in ultrasound images, we compared models trained with and without SAS using the well-benchmarked MedSAM [11] foundation model.

- SAS enhances model robustness and generalizability, particularly across out-of-distribution target structures and anatomical regions. By promoting learned invariances and balancing the training set, SAS generates more instances of small structures while preventing overfitting to larger organs. Extensive experiments in both data-scarce and data-abundant scenarios confirm the effectiveness of SAS in improving performance across a wide variety of anatomical structures and datasets.

Indeed, the recent success of large foundation models trained on enormous quantities of data prompts a critical question: Can we enhance medical imaging datasets by improving the sufficiency and data diversity of training examples through non-generative transformations? By generating large-scale datasets from a limited number of real images, this approach diminishes the need for human labeling efforts, while ensuring the process remains free from hallucinations.

## Methods

Pre-training a foundation model involves two key steps: collecting a large and diverse dataset, and training a base model that can later be adapted via transfer learning for downstream applications. In this work, we focus on the "last mile" of this process — fine-tuning a foundation model for the task of multi-organ segmentation in ultrasound imaging. To achieve this, we utilize a lightweight version of MedSAM, a model pre-trained on a large-scale medical imaging dataset comprising over 1.5 million images across 10 imaging modalities, including 94,450 ultrasound images. By leveraging this pre-trained foundation model, we harness the power of large-scale medical data without the computational cost and resource burden of training from scratch. Fine-tuning such a foundation model for downstream tasks often yields superior performance, provided sufficient training data is available.

### Ethics statement

This study was reviewed and approved by the GE HealthCare Institutional Review Board. The study involved retrospective analysis of medical imaging data. For internal datasets (Kidney, Gynecology, and Follicle), all data were fully de-identified prior to access, and the requirement for informed consent was waived by the Institutional Review Board due to the retrospective nature of the study and the use of de-identified data. For publicly available datasets used in this work, ethical approval and informed consent procedures were handled by the original dataset providers, in accordance with their respective institutional and regulatory requirements.

### Segment Anything Small (SAS) image transformations

The proposed method employs a dual transformation strategy to simulate various organ scales and appearances, as illustrated in Fig 1. Step 1 enhances the model's robustness to scale differences, which is particularly important for accurately

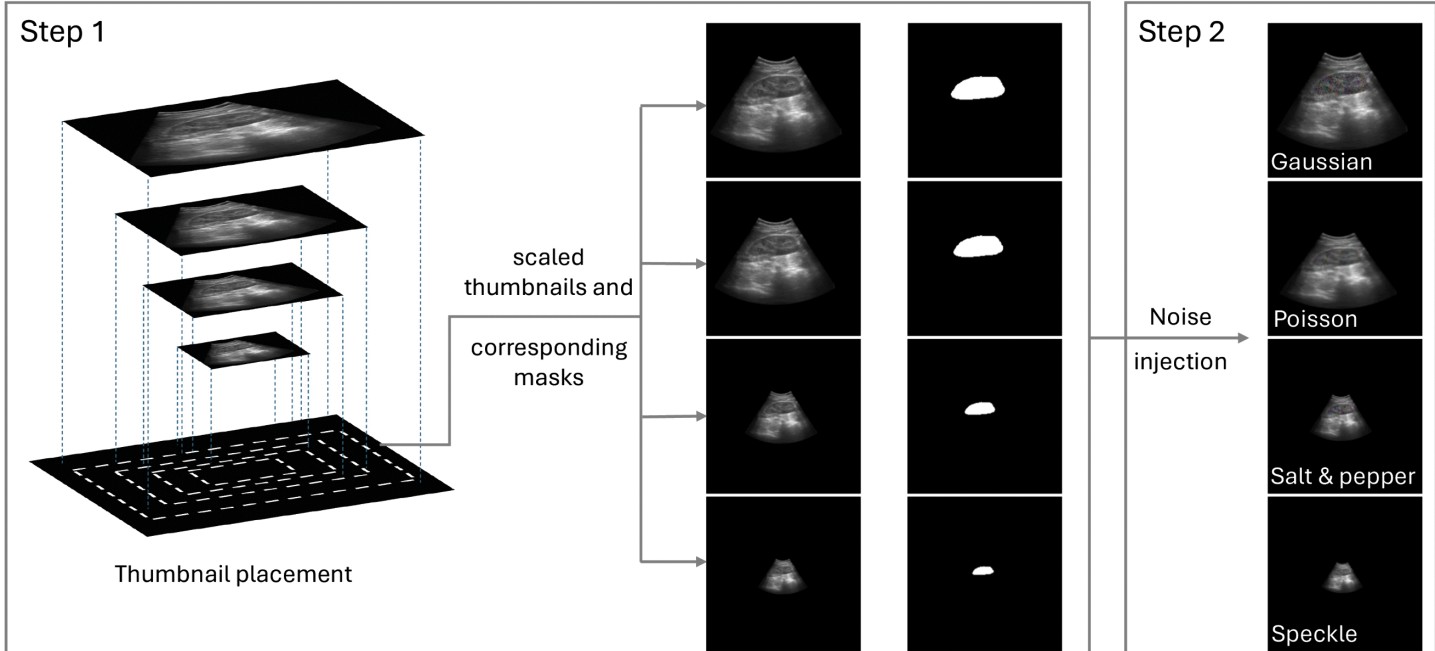

**Fig 1. SAS image transformations.** Step 1: Thumbnail generation by resizing images to simulate small anatomical structures. Step 2: Region-of-interest perturbations to represent variations in pixel intensity and noise levels.

segmenting small organs that may appear at different resolutions in real-world ultrasound images. Step 2, through noise injection, further enhances the model's ability to generalize across diverse tissue appearances while preserving the structural integrity of the original data. Together, these steps improve the model's performance and adaptability across a wide range of anatomical structures and imaging conditions.

Specifically, given a network input image and its corresponding mask pair $(x, y)$, where $x$ represents the image and $y$ denotes the mask, SAS generates a transformed image pair $(x', y')$ through a two-step process. First, the ultrasound window region within the image is resized down to generate scaled thumbnails, which are then placed on a black background of size equal to the desired network input size. Second, noise is injected into the organ region defined by the segmentation mask $y$ to simulate diverse textures. Next, we provide a detailed explanation of these steps.

**Step 1: Simulating organ scales.** In the first step, we simulate variations in organ size by extracting the region of interest (ROI) within the ultrasound window and rescaling it to create a thumbnail, as illustrated in Fig 1 (Step 1). This thumbnail is then placed on a black background with the same dimensions as the network's input image $x$, resulting in a scaled, zoomed-out representation of the organ. The black background is generated by creating a blank image matching the input dimensions with all pixel values set to zero, ensuring that no additional noise or artifacts are introduced. This operation is designed to mimic the on-screen display behavior of ultrasound systems during zoom in/out, where the ultrasound content occupies a subset of the screen, and the remaining pixels correspond to non-displayed regions outside the rendered ultrasound frame (e.g., outside the sector/convex/trapezoid viewport), which are typically encoded as zero-valued pixels. Using a zero-intensity background therefore preserves the display geometry without introducing artificial texture or speckle patterns. Because these padded regions contain no anatomical signal, they contribute minimal gradients during backpropagation and do not introduce spurious learning signals, keeping parameter updates driven by the displayed ultrasound content.

During fine-tuning, the input image size is fixed—for example, at 256 × 256 pixels in our selected architecture, as described in Section Data pre-processing. The thumbnail size is randomly selected during training, ranging from 64 × 64 to 256 × 256 pixels in this implementation, corresponding to a scaling ratio of 0.25 to 1. This simulates organs of varying sizes relative to the input image. Note that the input image size and thumbnail size can be adjusted depending on the neural network architecture.

**Step 2: Simulating diverse textures via noise injection.** In the second step, we simulate diverse textures in organs and tumors by injecting noise into the regions of interest (ROI) defined by the segmentation masks, as illustrated in Fig 1 (Step 2). The perturbation is achieved by applying one of several noise types—Speckle, Gaussian, Salt and Pepper, or Poisson—chosen at random with equal probability. Only one noise type is applied at a time to ensure controlled and realistic variations in texture.

During training, 50% of the images containing large structures, defined in the next section, are randomly transformed by SAS.

**Organ size definition.** We classify anatomy size based on its relative size to the original image: segmentation masks covering up to 3% of the image's original size are defined as "small structures," or "large structures" otherwise. This classification ensures that small organs or pathological structures (e.g., tumors) are not classified as small if they are zoomed in, i.e., if their actual size within the image exceeds the 3% threshold.

To distinguish small from large structures, we classify anatomy based on its relative size with respect to the full image. Specifically, structures whose segmentation masks occupy at most 3% of the image area are defined as small, while larger structures are categorized as large. This threshold was selected empirically based on the distribution of relative structure sizes in our datasets, which exhibits a strong concentration of samples below 3% area and a pronounced tail beyond this value (Fig 2). The 3% cutoff corresponds to a natural transition point in the size distribution, separating the dense cluster of small structures from larger, more variable anatomies.

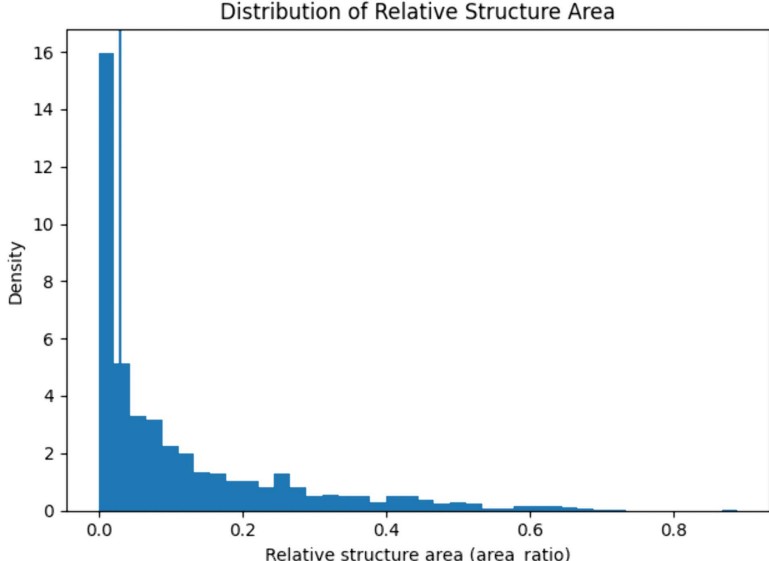

**Fig 2. Distribution of relative structure sizes used to define small and large anatomies.** Histogram of the relative area ratio (structure area divided by image area) across all datasets.

## Model training and selection

We adapt the network architecture described in [16,15], which includes a TinyViT image encoder [17], and a SAM-based prompt encoder and mask decoder [11,10]. The image encoder maps the input image into a high-dimensional embedding space, while the prompt encoder transforms user-click prompts (described in Section Multi-click prompts) into feature representations using positional encoding [18]. The mask decoder then fuses the image embeddings and prompt features through cross-attention mechanisms [19]. The SAS model is initialized using weights from a light-weight version of MedSAM [11], specifically LiteMedSAM [15]. LiteMedSAM was created by distilling a smaller image encoder, TinyViT, from the original ViT-based MedSAM image encoder, ensuring alignment in their image embedding outputs.

We then fine-tune the model using iterative click prompts, adapting MedSAM's prompt encoder to support a technique we call the "Multi-click prompting strategy," presented in Section Multi-click prompts. This strategy simulates realistic user-generated point prompts and is detailed in the following section. During fine-tuning, all trainable parameters in the image encoder, prompt encoder, and mask decoder are updated. The image encoder, prompt encoder, and mask decoder contain 5,726,740, 6,220, and 4,058,340 trainable parameters, respectively, resulting in a total model size of 9,791,300 parameters.

For optimization, we use a compound loss function combining Cross-Entropy loss for pixel-wise classification and Dice loss to improve segmentation accuracy [20]. The network is optimized using the AdamW optimizer [21] with hyperparameters $\beta_1 = 0.9$, $\beta_2 = 0.999$, an initial learning rate of $5e^{-5}$, and a weight decay of 0.01. Training is performed with a batch size of 8, on a single A100 (80 GB) GPU. Early stopping is applied if the combined loss on the validation set does not improve for 5 consecutive epochs. The final model checkpoint is selected based on the highest validation metric value.

**Multi-click prompts.** Point click prompts were used instead of bounding boxes for ultrasound segmentation, as bounding boxes are less effective for non-convex shapes and may inadvertently include unwanted anatomical structures, leading to incorrect segmentation. While previous research [22] suggests that box-based prompts can yield relatively higher segmentation accuracy, large bounding boxes may encompass multiple instances and background structures,

potentially confusing the model and resulting in incorrect segmentation. Other work [12] supports the use of iterative click prompts for achieving similar results to bounding boxes.

We implemented an iterative click prompt technique to simulate realistic user-generated interactions. Initially, the first point is selected from an area near the centroid of the reference standard (RS) mask. Specifically, the (x, y) coordinates are chosen from within the 30th percentile of the largest Euclidean distances to the mask boundary. Subsequent points are determined by calculating an error map between the model's prediction and the RS mask. Points are then selected from regions with the highest error. Each point is labeled as either a positive or negative prompt based on its location relative to the RS mask: a point is classified as a false negative if it falls inside the RS mask (indicating an error region), and as a false positive if it falls outside the RS mask.

## Data

The effectiveness of foundation models is strongly influenced by the size of the dataset, as highlighted by scaling law studies [11,23,24], particularly in medical imaging applications, where dataset availability and diversity can significantly impact the generalization capability of the models. For instance, MedSAM [11] - trained on over 1.5 million medical image-mask pairs across 10 imaging modalities, including 94,450 ultrasound (US) images - investigates the impact of varying the size of the training dataset, and confirms that increasing the number of training images significantly improves results in both internal and external validation sets. However, its evaluation on ultrasound has been limited to a single external dataset, focusing exclusively on fetal head segmentation—an easy-to-segment target where even the original SAM (without medical fine-tuning) achieved a Dice Similarity Coefficient (DSC) of 92.4% [11]. This underscores the need for a more comprehensive evaluation of segmentation performance on small structures across varied datasets.

In this work, we aim to investigate the model's ability to generalize effectively—leveraging the vast amounts of data seen during training to perform well on tasks involving unseen data, a core principle of foundation models. Specifically, we evaluate the performance of such models on out-of-distribution anatomies—specifically, i.e., datasets containing structures and imaging conditions not encountered during training. Additionally, we assess the feasibility and impact of the proposed SAS approach in improving segmentation performance for small anatomical structures, which are often challenging due to their size and variability.

To achieve this, we propose two evaluation scenarios:

1. **Scenario 1** investigates the model's ability to generalize to new, out-of-distribution anatomies using a limited training dataset consisting of 1,050 grayscale images from an internal dataset. These images depict both longitudinal and transverse views of renal anatomy, with each image being a single frame. The training data was randomly split per patient into 80% for training and 20% for validation, with the objective of segmenting the kidney.

2. **Scenario 2** examines the impact of increasing the training dataset size by incorporating a larger variety of ultrasound image data, consisting of 87,000 images of gynecological structures. As in Scenario 1, the data in Scenario 2 was split into 80% for training and 20% for validation, with the goal of segmenting these gynecological anatomical structures.

In both scenarios, we evaluate the same test sets, including small anatomies such as breast tumors, thyroid, nerve, vessels, gallbladder, and ovarian follicles. In Scenario 1, all these anatomies are considered out-of-distribution target structures, while in Scenario 2, only the anatomy of the ovarian follicle becomes part of the in-distribution target structure data.

The test set includes both an internal follicle dataset and external publicly available medical image segmentation datasets, obtained from the Cancer Imaging Archive [25] and Kaggle, as detailed in Table 1. These public datasets are well-established in the literature and include reference standard annotations provided by human experts. None of the holdout test datasets were ever used during model fine-tuning for either training or validation. In the following, we present each dataset and its role in model development. Table 1 summarizes patient demographics and key dataset characteristics.

**Table 1. Ultrasound datasets used for training and testing.**

| Dataset | Used for | Nb. masks | Unique patients | Object of interest | Demographics/clinical info |
|---|---|---|---|---|---|
| Kidney | train | 1,050 | 1,050 | Kidney | Not available |
| Gynecology | train | 80,000 | 920 | Uterus | Not available |
| Follicle | test | 69,000 | 71 | Follicle | Not available |
| Abdominal [26] | test | 42 | 11 | Kidney, vessels, gallbladder | 40% female, age = 27 ± 3 years; no history of abdominal pathology or known disease. |
| BUSI [27] | test | 665 | 600 | Breast benign and malignant tumors | 100% female, age 25–75 years; 454 benign and 211 malignant cases. |
| BrEaST [25] | test | 264 | 256 | Breast tumor | 100% female, age 18–87 years; normal, benign, and malignant categories. Histological diagnoses are reported in [25]. |
| DDTI [28] | test | 656 | 299 | Thyroid | 90% female, age = 57 ± 16 years; lesions include adenomas, thyroiditis, cystic nodules, and thyroid cancers. |
| Neck [29] | test | 2323 | N/A | Nerve structures | Not available. |

Nb. denotes number. N/A indicates not available.

**Data pre-processing.** The ultrasound region of interest was extracted from DICOM images, using the DICOM metadata and converted to PNG format. Next, the images were resized at the longest dimension to align with the input size of the model's encoder, i.e., 256 pixels, using bilinear interpolation. We then apply min-max normalization from 0 to 1, and pad the resized images with zero values, as needed, to create square dimensions of 256 × 256 × 3. The preprocessing for reference standard masks is similar, with a few key differences. Instead of bilinear interpolation, we use nearest-exact interpolation to preserve the mask's pixel values. After resizing the masks to match the input size of the image encoder, we pad them with zeros to achieve square dimensions of 256 × 256 × 3.

## Quantitative metrics

The *Dice Similarity Coefficient* (DSC) [30] and *Normalized Surface Distance* (NSD) [30], are widely used for evaluation in medical image segmentation tasks. Both metrics, range from the smallest value of 0 indicating the worst performance, to the highest value of 1.0 indicating perfect alignment. DSC is a measure of overlap between the predicted segmentation $\mathbf{P}$ and the reference standard segmentation $\mathbf{R}$. It is defined as:

$$\text{DSC} = \frac{2|\mathbf{P} \cap \mathbf{R}|}{|\mathbf{P}| + |\mathbf{R}|}$$

where $|\mathbf{P} \cap \mathbf{R}|$ represents the intersection of the predicted and reference standard segmentation, and $|\mathbf{P}|$ and $|\mathbf{R}|$ are the cardinalities of the predicted and reference standard sets, respectively.

The NSD is an uncertainty-aware segmentation metric that measures the overlap between two boundaries. It estimates which fraction of a segmentation boundary is correctly predicted with an additional threshold $\tau$ related to the clinically acceptable amount of deviation in pixels [31], and is defined as:

$$\text{NSD} = \frac{|\mathbf{S_P} \cap \beta_{\mathbf{R}}^{(\tau)}| + |\mathbf{S_R} \cap \beta_{\mathbf{P}}^{(\tau)}|}{|\mathbf{S_P}| + |\mathbf{S_R}|}$$

where $S_P$ is the boundary of the predicted segmentation $\mathbf{P}$, $\beta_{\mathbf{G}}^{(\tau)}$ is the border regions of $\mathbf{R}$ at tolerance $\tau$, $S_R$ is the boundary of the reference standard segmentation $\mathbf{R}$, $\beta_{\mathbf{P}}^{(\tau)}$ is the border regions of $\mathbf{P}$ at tolerance $\tau$, and $\tau$ is the degree of strictness for what constitutes a correct boundary.

Unlike DSC, which focuses on area overlap, NSD evaluates the accuracy of the segmentation boundaries, making it particularly sensitive to errors in the shape and contour of small objects [30]. It is thus a measure of what fraction of a segmentation boundary would have to be redrawn to correct for segmentation errors.

## Results

We conducted experiments in two scenarios, as detailed in Section Data. Scenario 1 uses a limited dataset of 1,050 renal ultrasound images to train the model, while Scenario 2 incorporates a larger dataset of 87,000 gynecological ultra-sound images to study the impact of dataset size and diversity. Both scenarios are evaluated on the same test set, which includes small anatomical structures such as breast tumors, thyroid, nerves, vessels, gallbladder, and ovarian follicles.

### Iterative click prompt

We begin by evaluating the effectiveness of iterative click prompts for segmentation performance. Iterative click prompts were generated as described in Section Multi-click prompts, and this strategy was applied during both training and inference. Performance was assessed in Scenario 1 on a holdout test split of the kidney dataset (n = 126) using Dice scores across the number of iterative prompts used, ranging from 1 to 10. Evaluations were conducted under two different fine-tuning scenarios for iterative click prompting: (a) full fine-tuning, where the image encoder, mask decoder, and prompt encoder were all fine-tuned, and (b) partial fine-tuning, where the image encoder was frozen while only the mask decoder and prompt encoder were fine-tuned. Additionally, a bounding box prompt approach with full fine-tuning was evaluated on the same dataset for comparison.

As shown in Fig 3, the iterative click prompts achieve comparable performance to bounding box prompts with just 2 click prompts, aligning with previous findings that iterative click prompts can surpass bounding boxes as the number of clicks increases [12]. The results also reveal a significant difference between full and partial fine-tuning. Full fine-tuning consistently achieved higher Dice scores with fewer prompts when compared to partial fine-tuning, and therefore, we adopt this setup for all subsequent experiments in this section. Notably, the full fine-tuning approach shows performance

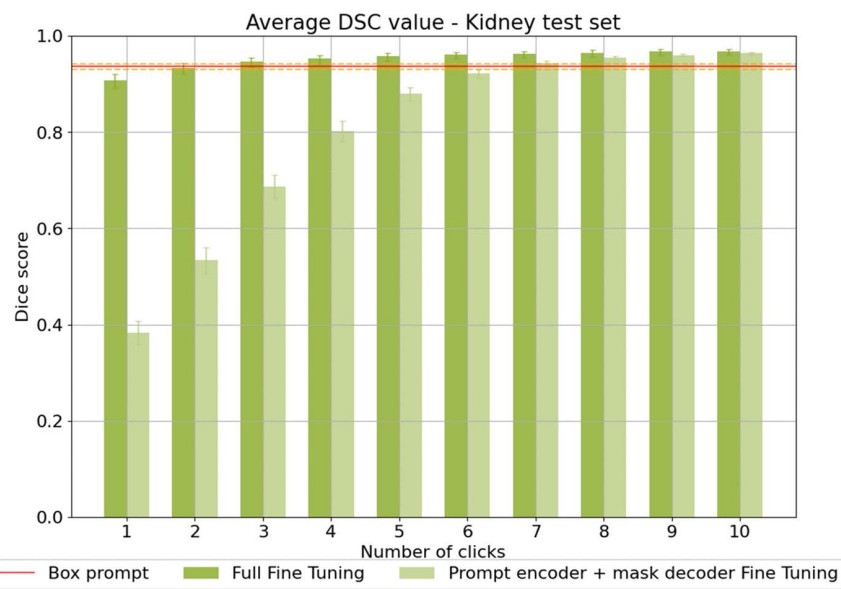

**Fig 3. Segmentation performance: iterative point vs. bounding box prompts.** Comparison of segmentation performance using iterative point and bounding box prompts with full and partial fine-tuning. Dice scores are reported for iterative point prompts ranging from 1 to 10.

comparable to bounding box prompts in as few as 2 iterative click prompts, highlighting its efficiency and effectiveness in achieving high-quality segmentation.

## Visualizing data diversity

To better understand the impact of SAS on data diversity, we computed feature embeddings using the ImageNet-1K [32] pre-trained VGG-16 model for both original ultrasound images and SAS-generated images. These embeddings were visualized using t-SNE plots in both Scenario 1 and Scenario 2, as shown in Fig 4. The plots reveal well-separated clusters for different test sets, underscoring the diversity of the evaluation datasets. In Scenario 1, where the original dataset is severely limited (1,050 renal ultrasound images), the original data points form a tight cluster (light orange), indicating restricted variability. SAS-generated data points (dark orange), created by applying SAS 20 times per image, occupy distinct regions of the data subspace, expanding the feature space, and improving the model's ability to generalize. In Scenario 2, due to computational constraints and for the sake of improving visualization, we randomly down-sampled the gynecological training dataset to 20,000 images and applied SAS once per image. Here, SAS-generated points also enhance data variety, though the effect is less pronounced compared to Scenario 1 due to the high diversity of the original data itself.

While the t-SNE plots provide meaningful insights into data diversity, it is important to note that these plots help us visualize data points reduced to two dimensions, so the plots may not fully capture the complexity of the data. A more complete picture regarding the effectiveness of SAS in enhancing data diversity can be obtained by also considering the improved generalization performance demonstrated in the next section.

## Quantitative performance evaluation of small structures segmentation

Building on the insights from the t-SNE analysis, which demonstrated SAS's ability to enhance data diversity and improve generalization, we now quantitatively evaluate its impact on segmenting anatomical structures in ultrasound. Figure 5 presents the Normalized Surface Distance (NSD) and Dice Similarity Coefficient (DSC) results across all six test datasets (detailed in Table 1) under two training scenarios: (a) Scenario 1 (orange lines), trained on a small, low-variety kidney image dataset, and (b) Scenario 2 (blue lines), trained on a large, high-variety gynecological image dataset. Performance

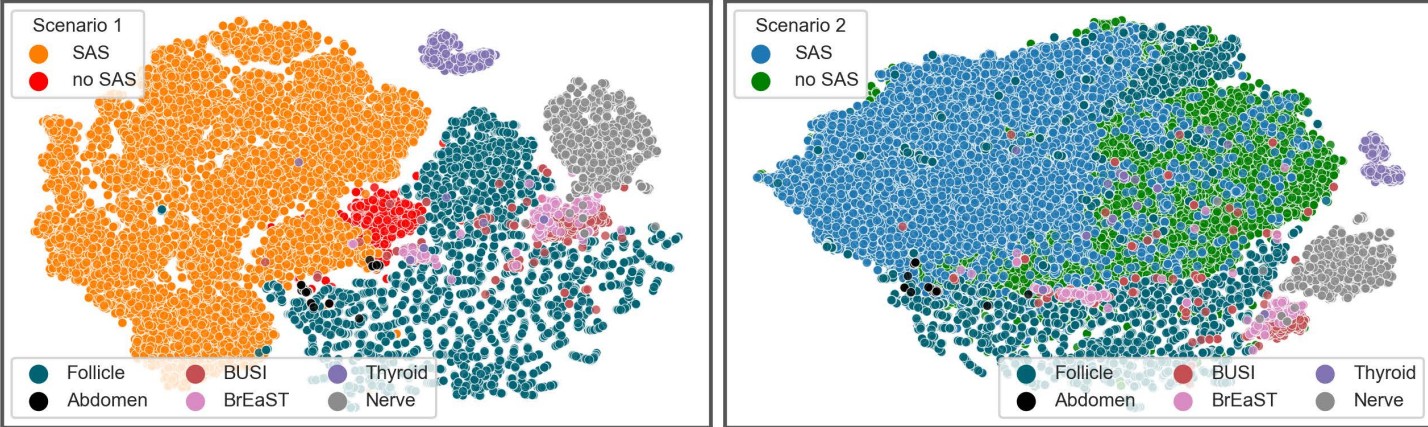

**Fig 4. Two-dimensional t-SNE visualization of feature embeddings for SAS-generated images, training set images, and test set images in Scenario 1 (left) and Scenario 2 (right).** SAS-generated images in both scenarios total 20,000, while non-SAS images in Scenario 2 were downsampled to 20,000 for comparison. Test set details are provided in Table 1. The embeddings were extracted using a VGG-16 encoder pre-trained on ImageNet-1K, with a t-SNE perplexity of 50.

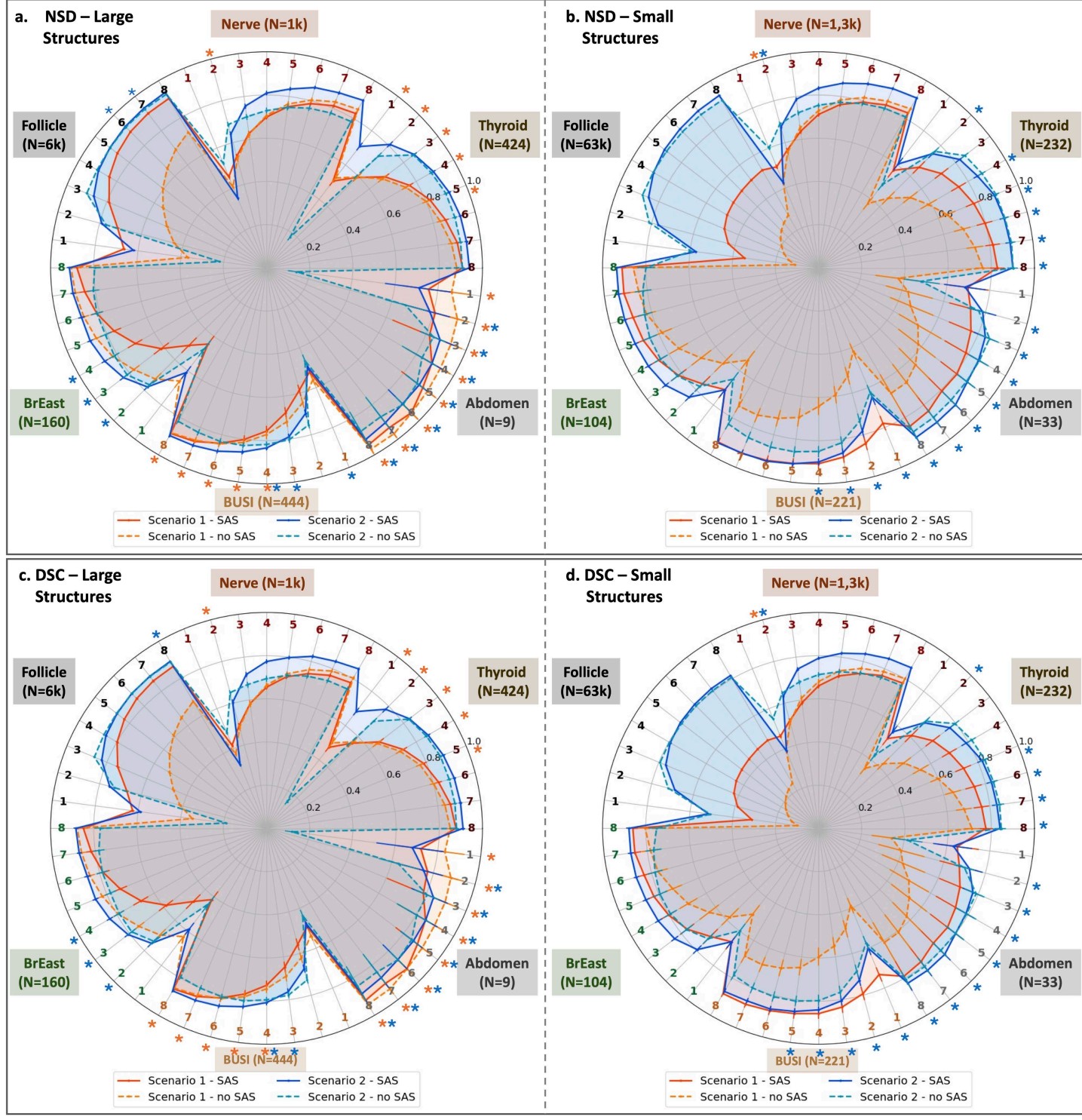

**Fig 5. Evaluation of SAS.** Normalized Surface Distance (NSD) and Dice Similarity Coefficient (DSC) results for all test datasets (see Table 1) in two scenarios: **a)** Scenario 1 (orange lines), using a low-variety training dataset of kidney images, and **b)** Scenario 2 (blue lines), using a high-variety training dataset of gynecological images. Results are shown for models trained with and without SAS. Numbers 1 to 8 indicate the number of iterative click prompts used in the segmentation process. Confidence intervals (CIs) were bootstrapped 10,000 times. Asterisks (*) indicate that baseline non-SAS and SAS are not significantly different.

is compared between models trained with and without SAS, highlighting its impact on segmentation accuracy across diverse anatomical structures. SAS consistently improves performance across both small and large organs, with particularly notable gains for small structures, where scale variations pose a greater challenge.

Notably, SAS in Scenario 1 (orange solid line) achieves performance levels comparable to the more diverse training set in Scenario 2 (blue solid line) for many test sets, demonstrating its ability to enhance generalization even with limited training data. Quantitatively, SAS improves average DSC by 0.084 [95% CI: 0.056, 0.111] in Scenario 1 and by 0.043 [95% CI: 0.022, 0.068] in Scenario 2, averaged across all test sets and eight click-prompt scenarios. When focusing on small organs, SAS achieves an average DSC improvement of 0.16 [95% CI: 0.132, 0.188] in Scenario 1 and 0.035 [95% CI: 0.018, 0.052] in Scenario 2, averaged across all test sets and eight click-prompt scenarios.

Across datasets, SAS demonstrates substantial improvements. For instance, in Fig 5(d), within the BUSI dataset, SAS achieves a DSC improvement of up to 0.35 (Scenario 1, one-click prompt), with an average DSC gain of 0.253 [95% CI: 0.216, 0.289] across eight click-prompt scenarios. The BrEaST dataset shows the largest DSC improvement of 0.161 for a single-click prompt, with an average increase of 0.101 [95% CI: 0.079, 0.125] over eight clicks. In the Abdomen dataset, SAS improves DSC by up to 0.307 for a single-click prompt, with an average gain of 0.234 [95% CI: 0.202, 0.268]. For the Thyroid dataset, the best improvement is observed with a two-click prompt (DSC increase of up to 0.185), with an average improvement of 0.133 [95% CI: 0.104, 0.16]. The Nerve dataset exhibits a maximum DSC improvement of 0.095, with an average DSC gain of 0.04 [95% CI: -0.02, 0.087]. The limited improvement observed on the nerve dataset is consistent with prior findings that nerve segmentation in ultrasound is primarily constrained by the need for long-range anatomical context and preservation of thin, deformable structures, which are not explicitly addressed by scale-based augmentation alone [33].

In the follicle test set, while SAS significantly boosts average DSC in Scenario 1 by 0.256 [95% 0.234, 0.258) points for small structures, it does not fully bridge the performance gap relative to Scenario 2 (blue lines). This suggests that although SAS enhances segmentation robustness, it cannot entirely compensate for the inherent limitations of the lower-diversity kidney dataset.

The violin plots in Figs 6–8 show the distribution of NSD and DSC metrics, grouping results from SAS and Non-SAS models while showing individual data points for all six test sets. As before, Scenario 1 is trained on a low-diversity dataset of kidney images, while Scenario 2 leverages a more diverse set of gynecological organs. Ideally, a generalizable segmentation model should achieve high performance with minimal labeled examples from the target anatomy, and these visualizations provide insight into how SAS influences segmentation robustness across different dataset compositions.

The Non-SAS model shows a higher concentration of low DSC and NSD values compared to the SAS model, indicating poorer segmentation quality, particularly for small anatomical structures. In most settings, the Non-SAS model achieves satisfactory performance only on larger organs (green dots in the plot), whereas SAS demonstrates strong segmentation improvements even with minimal user interaction.

The follicle dataset in Fig 6a, 6b serves as a representative case for small organs due to their inherently small size — illustrated by the red points in the violin plots — where 90% of the masks fall within our defined small-organ category, as detailed in Section Organ Size Definition. This enhancement brings performance closer to that observed in Scenario 2, which serves as a best-case benchmark and upper bound for the model's performance on the follicle dataset. As expected, when training is conducted in Scenario 2, the performance gap between SAS and Non-SAS narrows. However, SAS continues to provide added value, particularly with a limited number of interactive clicks.

## Qualitative performance evaluation of small structures segmentation

Figure 9 presents segmentation examples for three cases from the folicle testset, comparing the reference standard masks with predictions from both the SAS and non-SAS approaches using models trained in Scenario 1. The non-SAS approach struggles with small structures, often failing to segment them entirely, while also exhibiting under-segmentation

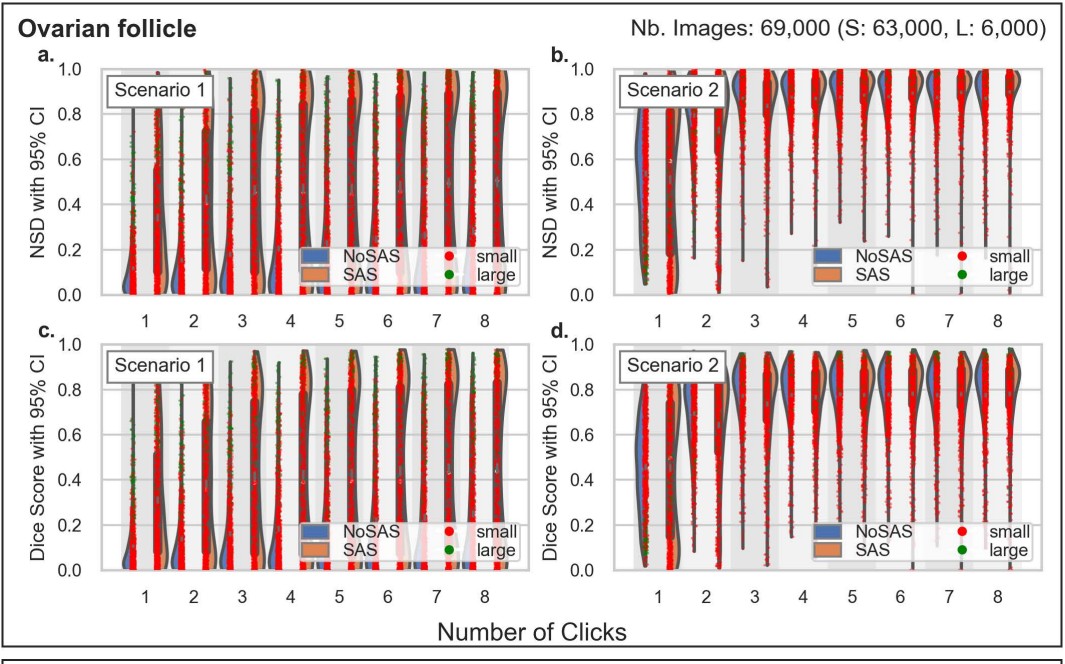

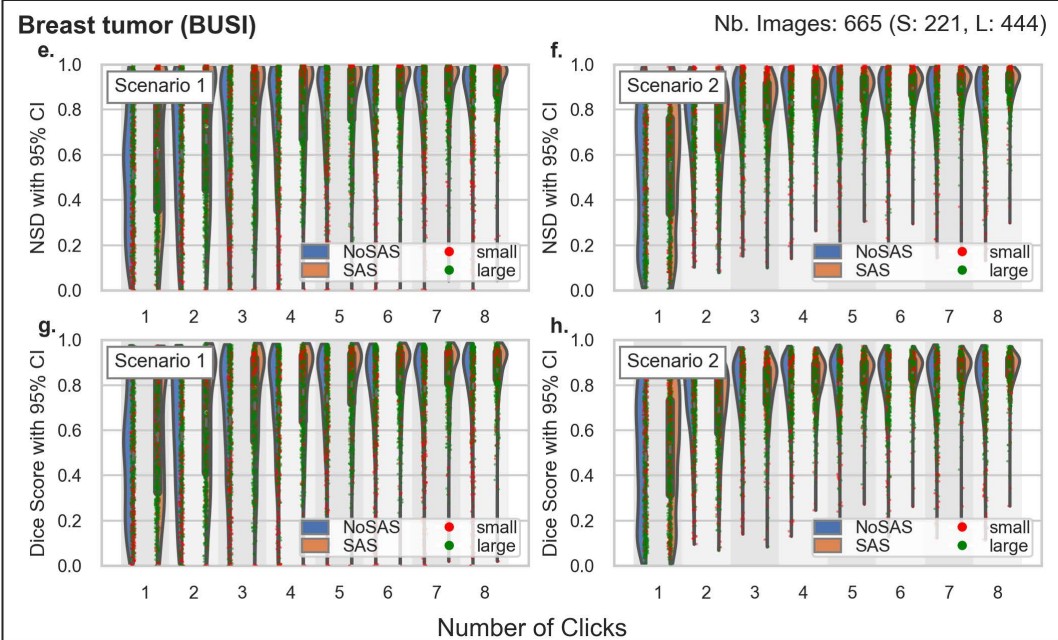

**Fig 6. Inference on Follicle and Breast Tumor - Violin plots showing the dice similarity coefficient and normalized surface distance scores for the small structures follicle and breast tumor datasets follicle (a-b) and BUSI (c-d) respectively.** The training dataset for Scenario 1 consists of kidney images, while Scenario 2 uses gynecological images.

in larger structures. In contrast, the SAS approach demonstrates improved accuracy across different anatomical sizes, effectively capturing small structures while also providing more complete segmentation for larger ones.

The quantitative results presented in Section Quantitative performance evaluation of small structures segmentation and qualitative results in Fig 9 highlight that, despite the limited fine-tuning dataset size and distinctly different structural and

PLOS Digital Health

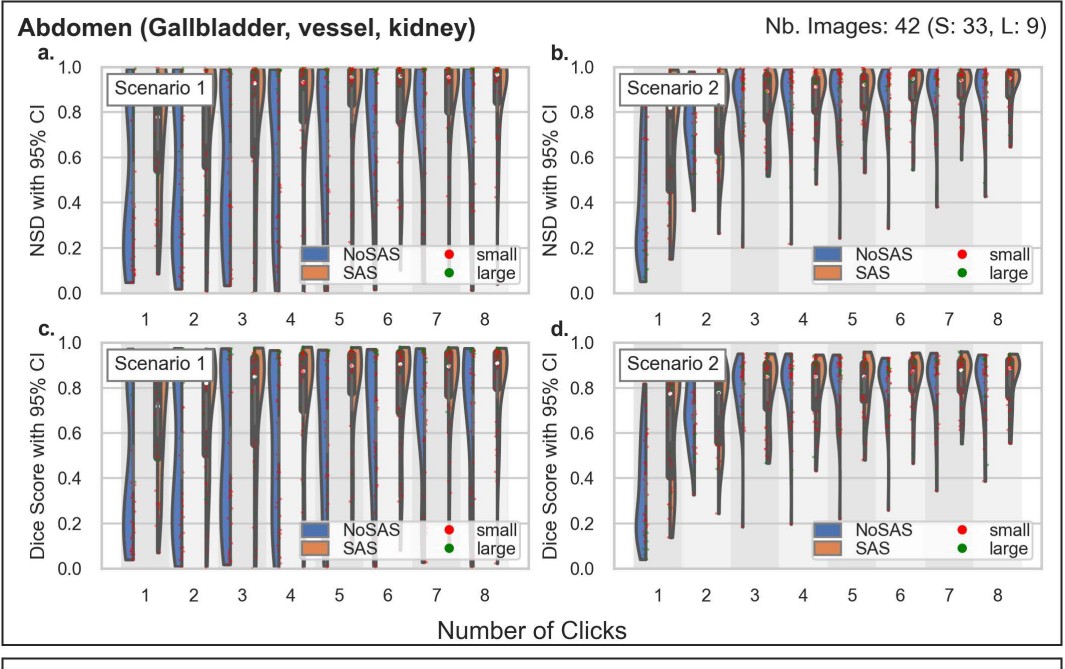

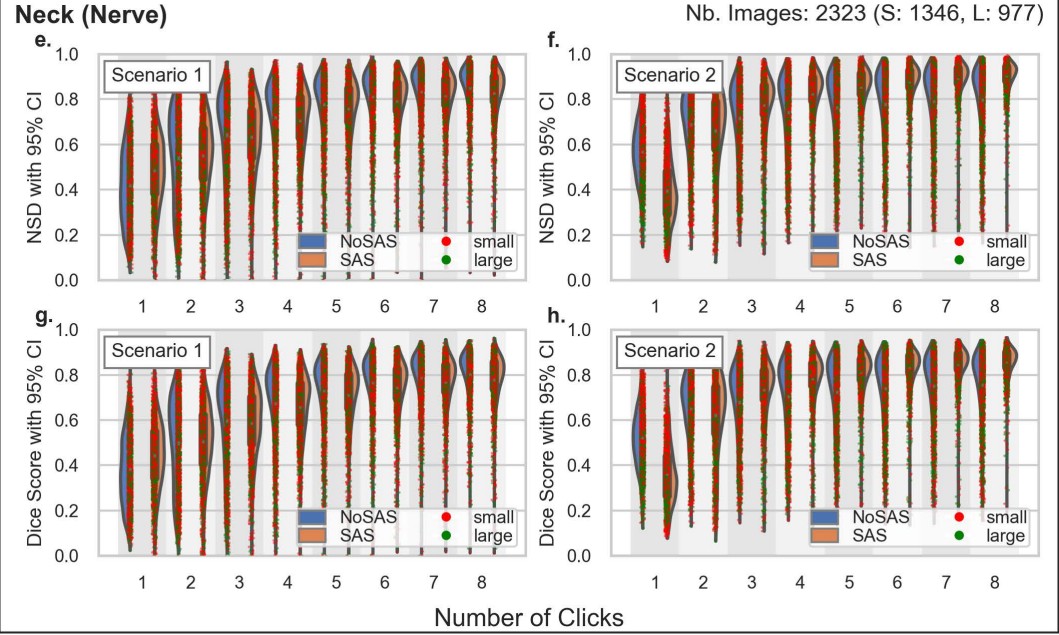

**Fig 7. Inference on Abdomen and Nerve - Violin plots showing the NSD and DSC scores for the Abdomen (a-b) and Neck (c-d) datasets.** The training dataset for Scenario 1 consists of kidney images, while Scenario 2 uses gynecological images.

anatomical characteristics in the test set, SAS effectively accomplishes the segmentation of small organs and structures, which also underscores the importance of methods for increasing data variety for developing robust models. The ability to generalize across various anatomical structures, despite being trained on a different dataset, demonstrates the adaptability and effectiveness the proposed method.

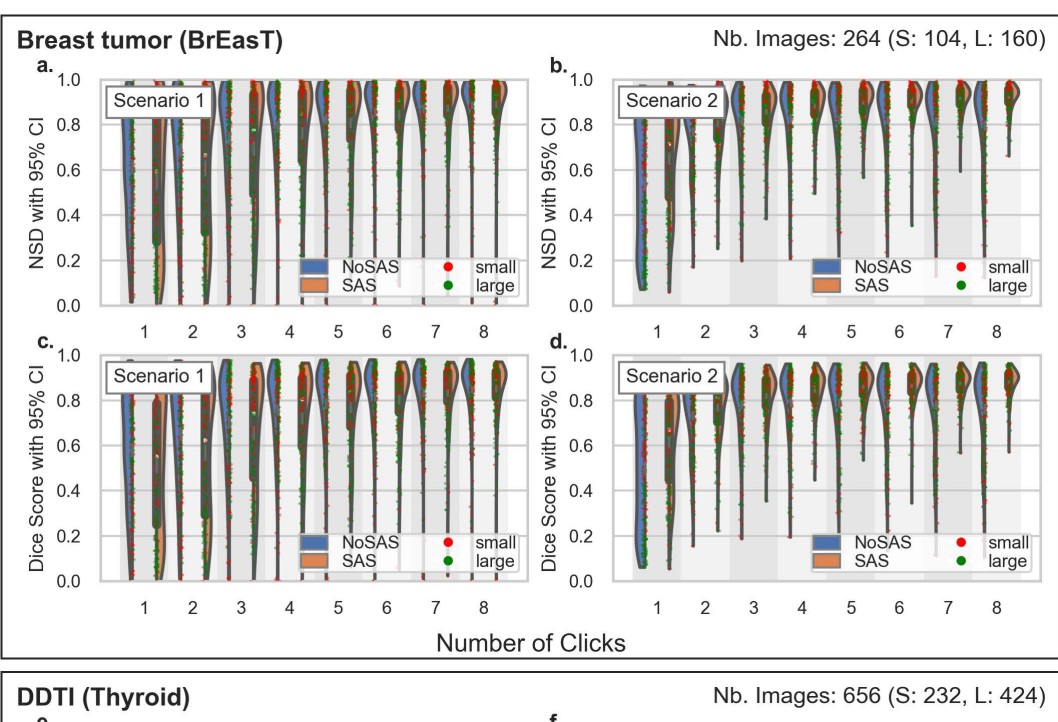

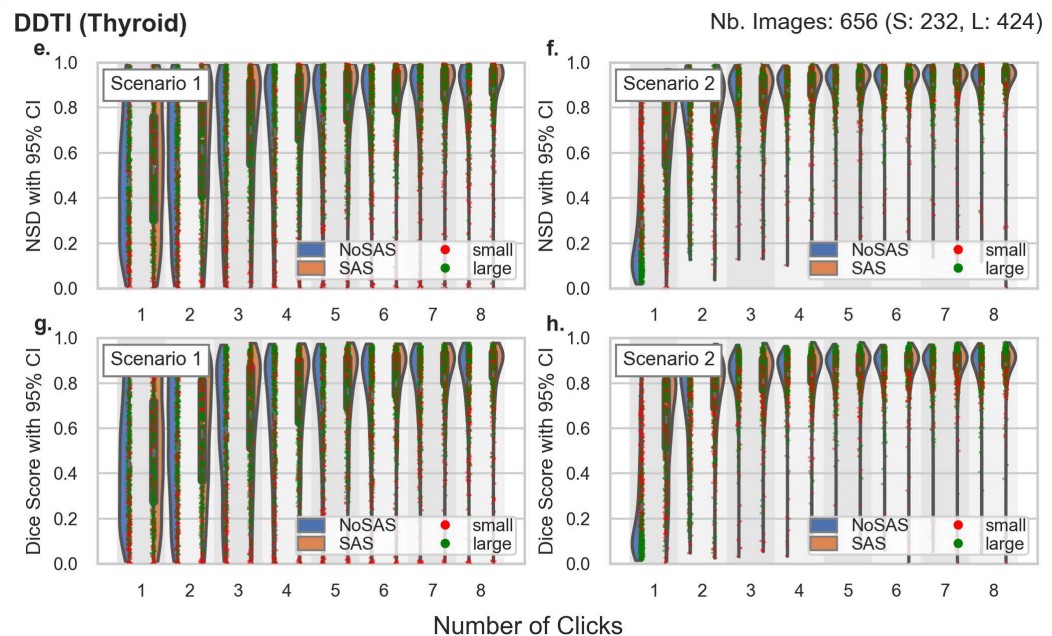

**Fig 8. Inference in BrEaST and Thyroid test sets.** Violin plots showing the DSC and NSD scores for the BrEaST (a-b) and DDTI (Thyroid) (c-d) datasets. The training dataset for Scenario 1 consists of kidney images, while Scenario 2 uses gynecological images.

## Ablation study: effect of scale and noise perturbations

To assess the individual and combined contributions of scale perturbation and noise perturbation in our SAS strategy, we conducted an ablation study in which the model was trained on the Kidney dataset and evaluated on the corresponding test sets. Performance is measured using the Dice coefficient as a function of the number of user interactions (1–8 clicks).

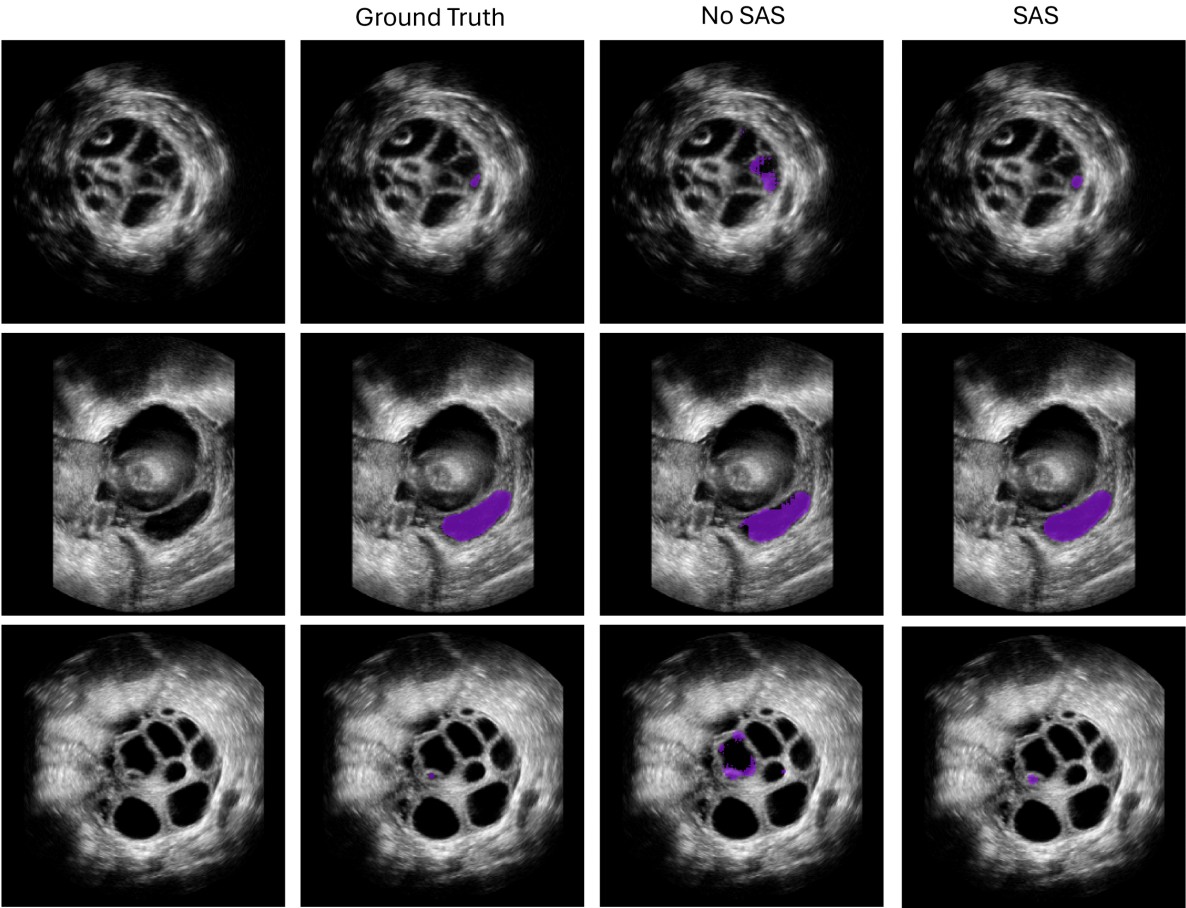

**Fig 9. Visualized segmentation examples of non-SAS and SAS segmentation masks on the follicle dataset.**

To further analyze the impact of SAS under different anatomical conditions, we stratify the results by object size into small and large organs. Figure 10 summarizes the results averaged across the test datasets. Each curve represents the mean Dice score, with shaded regions indicating the standard deviation across datasets.

For small structures (Fig 10, left), the baseline model exhibits limited improvement as the number of clicks increases, saturating at a relatively low Dice score. Introducing noise perturbation alone improves robustness during early interactions, while scale perturbation alone yields larger gains as additional clicks are provided. The full SAS strategy consistently outperforms all ablated variants across all interaction regimes, achieving both faster convergence at early clicks and higher final accuracy. For large structures (Fig 10, right), all methods benefit substantially from increased user interactions. Scale perturbation provides clear improvements over the baseline, especially at intermediate and later clicks, whereas noise perturbation alone yields more modest gains. The combined SAS strategy achieves the highest Dice scores across all interaction levels. These results highlight the complementary nature of scale and noise perturbations, particularly for challenging small anatomical targets.

## Discussion and conclusions

In this work, we address the challenge of accurate segmentation in Ultrasound imaging, particularly for small anatomical structures, where data scarcity and limited annotation quality often hinder the generalization of deep learning models.

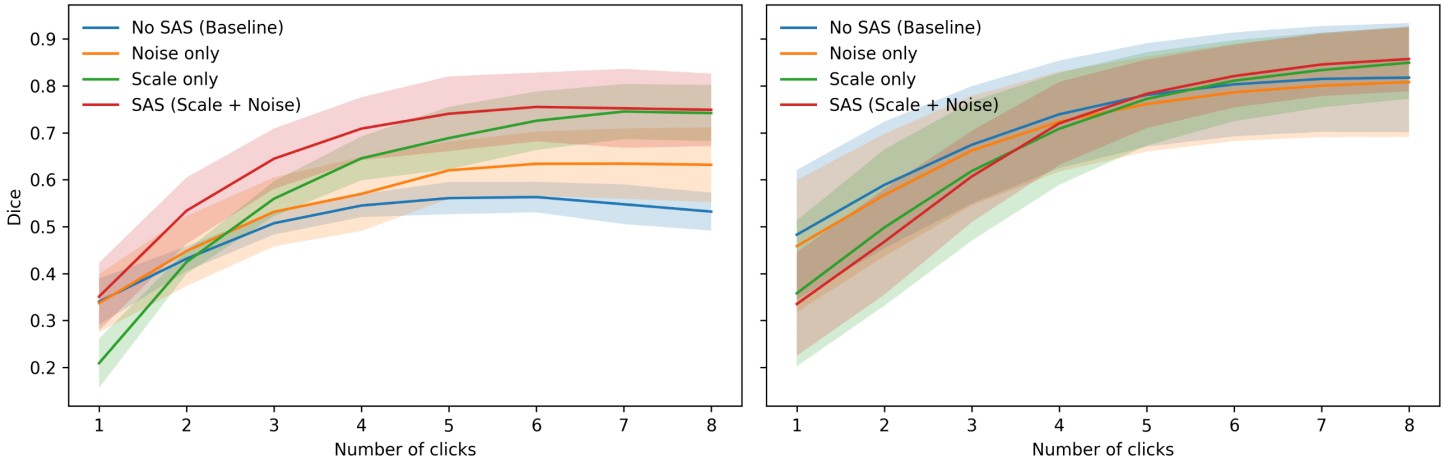

**Fig 10. Ablation of scale and noise perturbations in SAS.** Mean Dice score as a function of the number of user clicks (1–8), averaged across all test datasets for small (left) and large organs (right) for No SAS (baseline), Noise only, Scale only, and SAS (Scale + Noise). Shaded regions denote the standard deviation across datasets.

Segment Anything Small (SAS) is easily applicable to any US image, as it does not rely on external segmentations or separately trained generative models. Unlike generative methods, SAS avoids the introduction of artifacts, unwanted structures, or hallucinations, making it highly suitable for medical applications where accuracy and reliability are critical.

The proposed approach is based on the assumption that the "expected" domain shift for a specific anatomy can be simulated by applying SAS to a single source domain, enabling a deep model trained with SAS to better generalize to unseen domains. SAS demonstrates significant performance improvements across all six datasets - one internal and five external - for seven segmentation tasks, namely ovarian follicles, breast tumor, thyroid, gallbladder, vessels, kidney, and nerve. Despite being trained on a small, kidney-only dataset lacking geometric feature diversity (Scenario 1), SAS generalizes well to unseen organs. It performs consistently across datasets from multiple centers with diverse imaging vendors and protocols, highlighting its ability to adapt to variations in small organs.

In the Large Structures of Scenario 1, however, SAS underperforms for large breast tumors in the BrEaST dataset (Fig 5a, 5c), orange lines. The mean Dice score for large organs decreases from approximately 0.63 (no-SAS) to 0.41 (SAS) at one click, corresponding to a reduction of approximately 0.22 Dice. This gap persists across interaction levels, with differences of approximately 0.15 at three clicks and 0.10 at five clicks, before narrowing to about 0.03 at eight clicks. Similar trends are observed in NSD, indicating that degradation is systematic in both overlap- and boundary-based metrics rather than driven by isolated cases. This is likely due to SAS's inability to capture the morphological and anatomical irregularities of large tumors after altering the training data distribution with artificially generated small target samples. Additionally, the limited kidney dataset, which also involves a large organ, may have caused the non-SAS models to overfit to large organs, as these models do not benefit from the regularization introduced by SAS.

Adding more data, as in Scenario 2, seems to address the overfitting issue where SAS consistently proves beneficial regardless of training set size. When dataset variability is increased by expanding the dataset size by a factor of 80 in Scenario 2 (data abundance), the impact of SAS diminishes but remains significant compared to non-SAS, with the breast segmentation task showing the most notable improvement. We attribute this to two key phenomena: SAS promotes learned invariances that enhance generalization, and it balances the training set distribution by generating additional cases of small structures while avoiding overfitting to large organs. Through extensive experiments in both data-scarce

and data-abundant scenarios, we demonstrate that SAS improves robustness and generalizability across out-of-distribution organs and diverse datasets.

Future work will examine the applicability of SAS to additional imaging modalities and assess its integration into clinical workflows to evaluate feasibility and generalizability in real-world medical scenarios. Because increasing data diversity through augmentation does not necessarily lead to improved performance [8], future studies should systematically analyze the effect of augmentation strength by varying parameters such as image scaling, noise intensity, and the probability of applying perturbations during training. In addition, we will investigate the inclusion of elastic deformation-based augmentations to better capture the geometric variability of elongated, deformable structures, such as nerves, and assess whether such transformations complement scale-based augmentation. Finally, the behavior of SAS will be evaluated within self-supervised and semi-supervised learning frameworks to study its effectiveness under reduced annotation regimes.

## Supporting information

**S1 Fig. Graphical_abstract**
(PDF)

## Acknowledgments

The authors would like to acknowledge GE HealthCare for providing institutional support for this research. No external funding or specific grant was received for this study. A preliminary version of this work was published in the SPIE Proceedings of Medical Imaging [34].

## Author contributions

**Conceptualization:** Danielle Ferreira, Ahana Gangopadhyay, Ravi Soni.

**Data curation:** Danielle Ferreira, Ahana Gangopadhyay.

**Formal analysis:** Danielle Ferreira.

**Funding acquisition:** Ravi Soni, Gopal Avinash.

**Investigation:** Danielle Ferreira, Ravi Soni.

**Methodology:** Danielle Ferreira.

**Project administration:** Danielle Ferreira, Hsi-Ming Chang.

**Software:** Danielle Ferreira, Ahana Gangopadhyay.

**Supervision:** Ravi Soni.

**Validation:** Danielle Ferreira.

**Visualization:** Danielle Ferreira.

**Writing – original draft:** Danielle Ferreira.

**Writing – review & editing:** Danielle Ferreira, Ahana Gangopadhyay, Hsi-Ming Chang, Ravi Soni, Gopal Avinash.

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
