## [Decision Letter · Decision Letter 0]

23 Dec 2025

Response to Reviewers'. This file does not need to include responses to any formatting updates and technical items listed in the 'Journal Requirements' section below.'. This file does not need to include responses to any formatting updates and technical items listed in the 'Journal Requirements' section below.* A marked-up copy of your manuscript that highlights changes made to the original version. You should upload this as a separate file labeled 'Revised Manuscript with Track Changes'.'.* An unmarked version of your revised paper without tracked changes. You should upload this as a separate file labeled 'Manuscript'.'. If you would like to make changes to your financial disclosure, competing interests statement, or data availability statement, please make these updates within the submission form at the time of resubmission. Guidelines for resubmitting your figure files are available below the reviewer comments at the end of this letter. We look forward to receiving your revised manuscript. Kind regards, Ismini LourentzouSection EditorPLOS Digital Health Ismini LourentzouSection EditorPLOS Digital Health Leo Anthony CeliEditor-in-ChiefPLOS Digital Healthorcid.org/0000-0001-6712-6626  **Journal Requirements:** 

1. Please send a completed 'Competing Interests' statement, including any COIs declared by your co-authors. If you have no competing interests to declare, please state "The authors have declared that no competing interests exist". Otherwise please declare all competing interests beginning with the statement "I have read the journal's policy and the authors of this manuscript have the following competing interests:"

2. Please ensure that your Ethics Statement is available in its entirety at the beginning of your Methods section, under a subheading 'Ethics Statement'. It must include:

i) The name(s) of the Institutional Review Board(s) or Ethics Committee(s)

ii) The approval number(s), or a statement that approval was granted by the named board(s)

iii) (for human participants or donors) - A statement that formal consent was obtained (must state whether verbal/written) OR the reason consent was not obtained (e.g., anonymity).

3. We ask that a manuscript source file is provided at Revision. Please upload your manuscript file as a .doc, .docx, .rtf or .tex.

4. Please provide separate figure files in .tif or .eps format.

5. We note that you have included your Figures within the body of your manuscript. Please remove the Figures from the body of your manuscript and upload them as separate Figure files.

6. In the online submission form, you indicated that “The datasets used for performance evaluation are publicly accessible and are referenced in Table 1. Additional datasets used in this study, including Kidney, Gynecology, and Follicle, are not publicly available due to protected patient health information. For further inquiries, please contact the Corresponding Author.”.

3. Uploaded as supplementary information.

**Additional Editor Comments (if provided):****Reviewers' Comments:** Reviewer's Responses to Questions

**Comments to the Author**

1. Does this manuscript meet PLOS Digital Health’s publication criteria? Is the manuscript technically sound, and do the data support the conclusions? The manuscript must describe methodologically and ethically rigorous research with conclusions that are appropriately drawn based on the data presented.? Is the manuscript technically sound, and do the data support the conclusions? The manuscript must describe methodologically and ethically rigorous research with conclusions that are appropriately drawn based on the data presented.

Reviewer #1: Yes

Reviewer #2: Yes

2. Has the statistical analysis been performed appropriately and rigorously?

Reviewer #1: Yes

Reviewer #2: Yes

3. Have the authors made all data underlying the findings in their manuscript fully available (please refer to the Data Availability Statement at the start of the manuscript PDF file)?

The PLOS Data policy requires authors to make all data underlying the findings described in their manuscript fully available without restriction, with rare exception. The data should be provided as part of the manuscript or its supporting information, or deposited to a public repository. For example, in addition to summary statistics, the data points behind means, medians and variance measures should be available. If there are restrictions on publicly sharing data—e.g. participant privacy or use of data from a third party—those must be specified.requires authors to make all data underlying the findings described in their manuscript fully available without restriction, with rare exception. The data should be provided as part of the manuscript or its supporting information, or deposited to a public repository. For example, in addition to summary statistics, the data points behind means, medians and variance measures should be available. If there are restrictions on publicly sharing data—e.g. participant privacy or use of data from a third party—those must be specified.

Reviewer #1: Yes

Reviewer #2: Yes

4. Is the manuscript presented in an intelligible fashion and written in standard English?

Reviewer #1: Yes

Reviewer #2: Yes

Reviewer #1: This article investigates and proposes a method called SAS for data augmentation, with the research demonstrating that the SAS-based approach effectively enhances the performance of image segmentation, particularly showing greater effectiveness for smaller targets.

The overall structure of the article is clear and well-supported by evidence. However, there are a few specific areas where additional clarification or improvement could aid reader comprehension:

1. In Figure 3, Scenarios 1 and 2 use colors that are too similar to represent SAS and no-SAS samples, making comparison challenging. Since Section 3.2 specifically discusses how no-SAS samples cluster tightly in Scenario 1, using more distinct colors would better highlight the separation between SAS and no-SAS distributions as observed through t-SNE visualization.

2. The SAS method achieves data augmentation through two steps: image scaling and noise addition. While both steps contribute positively—scaling improves small target detection and noise enhances robustness—the combined effect shown in Figure 3 via t-SNE doesn't clarify whether the dimensionality reduction reflects the impact of scaling, noise addition, or both. A visualization comparing feature distributions after each step separately would better illustrate their individual contributions.

3. The four types of noise (Speckle, Gaussian, Salt and Pepper, Poisson) used in Step 2 do correspond to typical ultrasound noises. However, in Figure 1, the image added at the bottom of Step 2 appears to be a mosaic image rather than one of the specified noise types. This inconsistency should be reviewed by the authors.

The article makes significant contributions but these specific refinements would strengthen its communication of findings.

Reviewer #2: 1. phrases such as “emerges as a powerful tool” read as promotional; please revise to reflect performance association rather than definitive superiority.

2. Justify the 3% threshold for “small structures” (Methods, Section 2.1.3, lines 149–154): Please provide a citation or empirical rationale for this cutoff, or include a sensitivity analysis showing robustness to alternative thresholds.

3. Expand justification for black background placement (Methods, Section 2.1.1, lines 131–134): Please explain why a zero-intensity background best reflects ultrasound acquisition physics and whether this risks domain shift.

4. The manuscript would benefit from explicitly stating whether scale and noise components were tested independently, and where those results are reported

5. The explanation for underperformance on large breast tumors is reasonable, but please add a quantitative comparison to show the magnitude and consistency of degradation

6. The nerve dataset shows minimal improvement; please add a brief explanation of why SAS may be less effective for elongated, low-contrast structures.

7. Tighten speculative language in Future Work (Discussion, lines 415–423): Please distinguish clearly between planned experiments and hypotheses, avoiding language that implies expected success.

**Figure resubmission:** While revising your submission, we strongly recommend that you use PLOS’s NAAS tool (https://ngplosjournals.pagemajik.ai/artanalysis) to test your figure files. NAAS can convert your figure files to the TIFF file type and meet basic requirements (such as print size, resolution), or provide you with a report on issues that do not meet our requirements and that NAAS cannot fix.

**Reproducibility:** To enhance the reproducibility of your results, we recommend that authors of applicable studies deposit laboratory protocols in protocols.io, where a protocol can be assigned its own identifier (DOI) such that it can be cited independently in the future. Additionally, PLOS ONE offers an option to publish peer-reviewed clinical study protocols. Read more information on sharing protocols at https://plos.org/protocols?utm_medium=editorial-email&utm_source=authorletters&utm_campaign=protocols To enhance the reproducibility of your results, we recommend that authors of applicable studies deposit laboratory protocols in protocols.io, where a protocol can be assigned its own identifier (DOI) such that it can be cited independently in the future. Additionally, PLOS ONE offers an option to publish peer-reviewed clinical study protocols. Read more information on sharing protocols at https://plos.org/protocols?utm_medium=editorial-email&utm_source=authorletters&utm_campaign=protocols

---

## [Decision Letter · Decision Letter 1]

3 Mar 2026

Segment anything small for ultrasound: Enhancing segmentation with non-generative augmentation

PDIG-D-25-00562R1

Dear Ph.D Ferreira,

We are pleased to inform you that your manuscript 'Segment anything small for ultrasound: Enhancing segmentation with non-generative augmentation' has been provisionally accepted for publication in PLOS Digital Health.

Best regards,

Ismini Lourentzou

Section Editor

PLOS Digital Health

**Additional Editor Comments (if provided):**

**Reviewer Comments (if any, and for reference):**

Reviewer's Responses to Questions

**Comments to the Author**

Reviewer #1: All comments have been addressed

Reviewer #2: All comments have been addressed

publication criteria? Is the manuscript technically sound, and do the data support the conclusions? The manuscript must describe methodologically and ethically rigorous research with conclusions that are appropriately drawn based on the data presented.? Is the manuscript technically sound, and do the data support the conclusions? The manuscript must describe methodologically and ethically rigorous research with conclusions that are appropriately drawn based on the data presented.

Reviewer #1: Yes

Reviewer #2: Yes

3. Has the statistical analysis been performed appropriately and rigorously?

Reviewer #1: Yes

Reviewer #2: Yes

4. Have the authors made all data underlying the findings in their manuscript fully available (please refer to the Data Availability Statement at the start of the manuscript PDF file)?

The PLOS Data policy requires authors to make all data underlying the findings described in their manuscript fully available without restriction, with rare exception. The data should be provided as part of the manuscript or its supporting information, or deposited to a public repository. For example, in addition to summary statistics, the data points behind means, medians and variance measures should be available. If there are restrictions on publicly sharing data—e.g. participant privacy or use of data from a third party—those must be specified.requires authors to make all data underlying the findings described in their manuscript fully available without restriction, with rare exception. The data should be provided as part of the manuscript or its supporting information, or deposited to a public repository. For example, in addition to summary statistics, the data points behind means, medians and variance measures should be available. If there are restrictions on publicly sharing data—e.g. participant privacy or use of data from a third party—those must be specified.

Reviewer #1: Yes

Reviewer #2: Yes

5. Is the manuscript presented in an intelligible fashion and written in standard English?

Reviewer #1: Yes

Reviewer #2: Yes

Reviewer #1: Thank you the authors for their careful and thorough responses to all points raised in the first round of review and for implementing the corresponding revisions to the manuscript.

All three specific concerns I raised have been effectively addressed:

1.Figure Visualization: The updated figure (now Figure 4) uses more distinct colors, significantly improving the visual contrast between SAS and no-SAS sample distributions and making the results clearer.

2.Method Contribution Analysis: The newly added Section 3.5 (Ablation Study) and Figure 10 explicitly disentangle and demonstrate the individual and combined contributions of the scale and noise perturbations through quantitative experiments, providing a strong response to the query regarding mechanistic explanation.

3.Figure Consistency: The noise examples in Step 2 of Figure 1 have been corrected to show the four actual noise types described in the text, resolving the previous ambiguity.

Upon review, the authors have sufficiently addressed and incorporated all revision suggestions from the previous round. The manuscript has seen substantial improvements in methodological explanation, experimental validation, and presentation of results. I find the revised manuscript meets the criteria for publication.

Recommendation: Accept for publication.

Reviewer #2: The new ablation study in Section 3.5 is a great addition, as it clearly breaks down how scale and noise perturbations each contribute to the model’s success.

Using a 3% area threshold to define "small structures" is now well-supported by the empirical data you've added to Figure 2.

The added transparency regarding the lower performance on large breast tumors and elongated nerves provides a more balanced view of the method's current limits.

Your explanation that the black background simulates actual ultrasound zoom behavior effectively clears up concerns about domain shifts.

Over all its a good manuscript. all the best.
